# Farmer's Perception, Agricultural Subsidies, and Adoption of Sustainable Agricultural Practices: A Case from Mongolia

**Buyannemekh Puntsagdorj ***, **Dulamragchaa Orosoo, Xuexi Huo and Xianli Xia**

College of Economics and Management, Northwest A&F University, Xianyang 712000, China; dulamragchaa.o@muls.edu.mn (D.O.); xuexihuo@nwafu.edu.cn (X.H.); xnxxli@163.com (X.X.)
* Correspondence: buyannemekh@muls.edu.mn; Tel.: +86-976-8802-9365

**Abstract:** The farmers' sustainable production behavior is viewed as the frontline measure that accomplishes sustainable development in agriculture. Finding ways to support farmers' adoption of sustainable agriculture practices (SAP) has become an issue of concern for researchers and policymakers. The paper aimed to investigate the impact of the current subsidy policy and other key variables on the adoption behavior of the Mongolian wheat growers. The generalized structural equation modeling was employed along with the protection motivation theory framework. The results show that the farmers who perceive high severity and vulnerability of soil erosion are more likely to adopt the SAPs. Moreover, the perceived efficacy of the practices and the farmers' perceived self competency contribute to the decision. The information and training are positively associated with adoption. We also reveal differences between the regions on adoption. Soil fertility has a significant negative impact. Finally, government subsidies are found to have no effect as these subsidies are not intended to promote sustainability. The study findings suggest that increasing farmers' awareness of the harmful effects of growth-oriented production practices, giving related information, and providing training and resources for the use of SAPs that are appropriate to the specific region. The results have implications for developing a policy targeted to promote the adoption of SAPs.

**Keywords:** farming practices; economic incentive; sustainable behavior; protection motivation theory; adoption decision; GSEM

## 1. Introduction

Sustainable development means providing current needs without compromising the ability of future generations to meet their own needs. As for the crop production sector, which is the main source of the provision for the population, a sustainable development foundation depends on the long-term fertility and productivity of the soil [1]. Unfortunately, a large amount of land has degraded beyond reclamation in the last four decades, while a few percent of these lands are rehabilitated during that period [2,3]. Soil degradation implies a long-term decline in soil fertility, soil erosion, or adverse changes in the physical and chemical properties of soil. Among the possible causes, soil erosion has been the major process that leads to degradation [4,5]. As stated in [6], the soil erosion rate has been much faster than the soil formation process and it has been found to be 100–1000 times higher for arable land than the natural background erosion rate.

The agricultural practices directly affect whether the soil erodes or retains fertile and provides long-term sustainable benefits. Contemporary agriculture has been dominated in many developing countries to solve economic and social issues by increasing agricultural productivity [7,8]. However, these growth-oriented agricultural practices have solved the challenge of providing agricultural production growth, leading to a decline in soil fertility (due to nutritional degradation, physical erosion, and loss of organic matter) [9]. The negative impacts of unsustainable farming are not only limited by harming the environment but also affects the productivity and profitability of the production in the long

run. Adoption and diffusion of sustainable agriculture practices (SAP) are considered to be the solution to cope with this unsustainable trend [8–10]. SAPs are viewed as a win-win strategy because of their potential to simultaneously address environmental, economic, and social issues [11]. In addition to becoming a protection against environmental damage [10,12], many studies have found that efficient usage of SAPs helps farmers to improve productivity and income [13] and contributes to reducing poverty and hunger [9,14]. Countries adopt different types of SAPs and governments encourage the adoption process through their effective policy measures [12,15]. However, farmers' adoption rate is still low in many countries [2,12,16]. Thus, explaining this phenomenon and getting a better insight into farmers' adoption behavior has become an issue of concern for researchers and policymakers.

Most of the arable area in Mongolia consists of brown loam soil with low humus and loose structure that is very vulnerable to erosion and degradation [17]. The soil degradation in Mongolia is mainly due to soil erosion. In addition to the soil feature, strong and sustained wind, the absence of natural wind barriers, and farmers' soil eroding conventional technology increase the soil vulnerability [18,19]. Around 46.5% of the total farming field were more-or-less eroded in 1990, but the amount of eroded land increased, and by 2010, all arable land was eroded to some extent [18]. Moreover, 60.6% of the analyzed fields had a major erosion at that time [20]. Like the other developing countries, the Mongolian agricultural sector has been directed to increase production and provide self-sufficiency in the past and now is transitioning from intensive to sustainable agriculture.

According to the Food Supply Law and the National Security Law, wheat is the main food staple and a strategic product in Mongolia. Although only one percent (1.35 million hectares) of the total land (1565 million hectares) is suitable for crop cultivation [21], it is possible to grow wheat, potatoes, and some vegetables to provide domestic demand. In the last five years, on average, 75% of the total sown area or 342.0 thousand hectares was planted with wheat, and about 360 thousand tons of wheat was harvested, which meets around 95 percent of the total domestic demand. Yield per unit area has increased significantly over the last decade, with an average annual yield of 1.27 ha, the highest average yield in wheat production development (The beginning of the development of modern wheat production in Mongola is considered to be from 1960, and the period up to now can be divided into 5 stages of development. In the previous four stages, the average yield was 0.77 tons/ha; 0.9 tons/ha; 1.2 tons/ha; and 0.79 tons/ha, respectively). As of 2019, a total of 98,000 people were employed in the crop production sector, of which 40% was related to wheat cultivation [22]. Crop production accounts for 17% ($302 million) of total agricultural output, which is sourced from mostly wheat [22].

Because of its importance, the wheat production sector is operating under huge government interventions. The Government of Mongolia (GoM) implements a specific policy that is dedicated to increasing wheat production. The policy has mainly consisted of various subsidies for the farmers to increase their operation scale (by encouraging them to cultivate more land), increasing productivity, and decreasing their financial burden [23]. However, the aspects of sustainable farming have not been reflected in government policy so far. Besides the Government production growth-oriented policy, the German-Mongolian joint project on sustainable agriculture (abbreviated as DMKNL in German) has been implemented since 2013. As stated by DMKNL, the yield per area has been steadily increasing; however, due to the factors such as unsuitable tillage, cultivation of the mono-crop, and inappropriate fertilizing, the soil has been degrading year by year [24]. Nyamsambuu and Ikhbayar [18] stated that, as of 2016, 25.4 and 213.6 tons/ha fine-grained fertile soils were blowing away from a slight and severe eroded area, respectively. The loss of humus was 4.6–6.8 tons/ha from slightly eroded soils, while it was 21.6–40.8 tons/ha for severely eroded areas. According to the DMKNL project report in 2016, the current adoption level of the SAPs is not sufficient [25]. Currently, Mongolia is in a period of transitioning from intensive to sustainable agriculture at the Government policy level (As a first step, "Land 4-Sustainable crop production development campaign" is developed and approved in the

beginning of 2020. Before, "Land 3 campaign" had been implemented since 2008 and was dedicated to provide the self sufficiency of wheat demand). Therefore, it is required to assess the current subsidy policy impacts on sustainable behavior as well as to study the determinants of the farmers' sustainable behavior.

Numerous studies on adoption decisions in agriculture have been carried out worldwide [26–28]. Early studies had employed conventional explanatory factors that are socio-economic characteristics of farmers and other exogenous factors such as environmental and macro-economic features. Many of these variables have been criticized because they were found to be not significant in many studies [29]. In recent literature, with the arising use of behavioral analysis in economic studies, the researchers have started to consider the psychological factors to explain the farmers' adoption behavior [13,30,31]. Studies have highlighted that including psychological factors and appropriately assimilating with the socio-economic factors would provide a better understanding of farmers' behavior in adopting SAPs [28,31,32]. For instance, as stated in [12], farmers might not adopt the SAPs even when the economic theory predicts they should. Moreover, studies suggest that taking multi-dimensional considerations into account would explain the complexity of decision-making on adoption [33,34].

The present study investigates the adoption of SAPs and examines the various socio-economic and psychological factors affecting wheat farmers' adoption decisions in Mongolia. In this regard, the study sets two objectives: (1) revealing the impact of current subsidy policy on the adoption decision to get an insight to a certain extent on the consistency of current policy with the sustainable development in agriculture, (2) investigating the various factors affecting farmers' adoption decisions of SAP. To do so, first, we aimed to analyze the use of several SAPs that are most common in Mongolia, such as minimum tillage, multi-crop rotation system, use of compost and manure, and straw mulching. Using the protection motivation theory (PMT), one of the common behavioral models on adoption decision and other factors from the multiple disciplines, we aim to explain the farmers' adoption intensity of the SAPs. This study contributes to the current literature as follows. To our knowledge, analysis on the adoption of SAPs and predicting the adoption decision has not been studied for the Mongolian context. Therefore, the paper would fill this gap and become one of the fundamental sources for both policymakers and later researchers. To promote sustainable farming, designing or modifying policies that address the barriers to sustainable farming can be a possible measure worth taking. Second, the study contributes to the literature on the adoption of SAPs that takes multi-dimensional factors into account. Specifically, proposing context suitable adoption models and developing an appropriate model specification that fits the sample data characteristics can provide improved and repeatable study material. Finally, the explaining power of the protection motivation theory on farmers' adoption decision of SAPs is tested in the Mongolian wheat farming sector.

## 2. Adoption of SAPs and the Government Policy Overview

Countries use a variety of agricultural practices for sustainable agricultural development [26,35]. For instance, Wezel, et al. [36] distinguished 15 categories of agroecological practices for sustainable agriculture, while Nasir Ahmad et al. [37] discerned 11 subthemes of SAPs (soil erosion control practices in the stud) observed in Asia. The benefits and impact of each practice vary depending on the features of the local area and the characteristics of that certain SAP. In general, according to [38] as cited in [27] (p. 2), the common attributes for SAPs are (1) conserving resources, (2) environmentally non-degrading, (3) technically appropriate, (4) economically viable, and (5) socially acceptable. The SAPs considered in this study are the most common practices among wheat farmers in Mongolia. Those are (1) multi-crop rotation system, (2) minimum tillage, (3) use of compost and manure, and (4) straw mulching.

Multi-crop rotation is a technique of growing different crops on the same land by recurring sequence over time to enhance soil fertility and plant protection [39]. In Mongolia, farmers often use the conventional wheat-fallow short-rotation system [17]. There is an

advantage that wheat production will be stabilized through this short system; however, productivity will be reduced due to loss of soil fertility in the long run. Later rehabilitation of degraded soils will require high costs, time, and effort. A small percentage of farmers use multi-crop rotation systems, such as three-field (wheat-fallow-potato; wheat-fallow-forage; wheat-fallow-rapes, etc.) or four-field systems.

Farmers have begun to shift from traditional soil processing with high mechanical operations to technologies that reduce tillage, but the transition has been slow [40]. Performing several technological operations in one pass or using fertilizers for soil processing, the reduced tillage or no-till system are the techniques for sustainable soil processing behavior. The reduced processing of the soil declines the vulnerability for erosion.

The use of manure involves the application of livestock waste, while the use of compost refers to the fine stabilized organic matter resulting from composting processes. These organic fertilizations reduce soil erosion rate, improve soil structure, and substitute the chemical fertilizers [36,37]. The mulching technique helps to prevent soil erosion, maintain humidity and heat. According to several studies in Mongolia, the moisture content of the straw-covered area was 9–14 mm higher than that of the unpaved area, and the humus content increased by 1.0–2.4% [41].

Mongolia has introduced various types of subsidies in 2008 to restore the previously declining wheat production. During the democratic revolution in 1990, the former state-owned farms disintegrated and were privatized. The former extensive government control and support /unified policy, loans, and grants/ was removed, and most of the farms had fallen into the hands of non-professional individuals. As in many other countries, this democratic shift significantly impacted agricultural production, leading to a sharp decline in the size of arable land and harvested crops. As a consequence, the average wheat yield had reached 0.79 tons/ha between 1990 and 2008, which is the all-time lowest level [23].

Subsidies include an output-based cash payment, soft loans for purchasing inputs, and a soft loan for purchasing agricultural machinery. As of 2019, around $25 was given per tons of wheat that met the domestic wheat quality standard. In terms of soft loans, only the farmers who purchase inputs and machinery from the Crop Supporting Fund (the impelementing agency of the Ministry of Food, Agriculture and Light Industry of Mongolia) are eligible to take the loans. For soft loans, it is not the whole amount considered, but the interest rate concessions are accounted as a subsidy. The soft loan for purchasing inputs has no interest, and farmers can repay their loan with their harvest. In the machinery soft loan case, the annual interest rate is 2% at maximum, while the market interest rate for the same loan is around 24% annually. According to the budget report of the Ministry of Food, Agriculture, and Light Industry of Mongolia, on average, 63–68% of the budget approved for crop production over the last five years has been spent on cash incentives and other subsidies for wheat. Moreover, according to the estimation of [23], the wheat production had the most share (38%) in total agricultural subsidy (other subsidies for agriculture include, the subsidies for wool and cashmere, meat, and milk), which was around $37 million.

In general, these subsidies are intended to increase the farmers' income, decrease their financial burden. The direct outcome that the Government expected is the increase in production and yield per area [40]. Consequently, the self-sufficiency ratio rose from 25% in 2008 to 80% in 2018 [42]. As reported in the World Bank review, the subsidy policy contributed to the farmers' financial capacity and increased production [23]. However, the impact on sustainability has not been studied so far.

## 3. Conceptual Framework and Research Hypothesis

In the previous literature, adoption behavior has been commonly considered as a dichotomous decision and treated as a binary variable for each specific practice [9,13,14,30]. However, our study proposed to take the intensity of adoption as a dependent variable. Several studies use the number of sustainable practices that farmers have employed during a specified period, which means as a count dependent variable [8,43,44]. Sharma et al. [45]

demonstrated that it is more appropriate to model the adoption of conservation agriculture as a multiple technology selection. Thus, understanding the decision of adoption intensity has become an important issue.

Mongolia's current wheat subsidies are more focused on increasing production and revitalizing the sector. Although not directly related to SAPs, these subsidies can be a motivator for the use of sustainable practices that reduce soil erosion to increase productivity and maintain long-term efficiency. Government supports can motivate the farmers to work efficiently and increase their productivity [46,47], and one way to increase the efficiency is to transform into sustainable farming [15]. From here, the first hypothesis we make in this study is:

**Hypothesis 1 (H1).** *Current agricultural subsidies have a positive impact on the SAP adoption.*

In terms of psychological works in this area, research has been widely based on the theory of planned behavior (TPB) [8,32,33], technology adoption model [10,48], and the protection motivation theory (PMT) [7,30,49]. These models use different psychological determinants to explain an individual's adoption behavior.

Of these, the theoretical framework of PMT is considered more consistent with the nature of SAP. SAP itself seeks to reduce future risks by applying appropriate practices to cultivation, while PMT theory links the perception of the risk and the perception of the possibility of overcoming the threat through engaging specific activity to explain the adoption decision. Furthermore, PMT employs a broader set of predictors than the theories mentioned above that may enhance our understanding of factors driving the adoption decision [31].

According to PMT, individuals facing a potential risk make two appraisals, threat appraisal and coping appraisal. In threat appraisal, individuals make perceptions on the severity of the threat and the vulnerability to the threat. If these perceptions are high, it is more likely that people will engage in protecting measures [30]. However, the awareness or perception of a threat alone does not encourage taking protective behavior [8,35]. The coping appraisal consists of perceived self-efficacy and perceived response efficacy. Self-efficacy refers to the perception of themselves are competent to conduct the actions, while the response efficacy relates to the perception of whether the actions could be effective [31].

As employed in the PMT in this study framework, we assumed that soil erosion endangers the productivity of the farmer in the long-term, the perception of threat encourages taking actions to repair or prevent further damage. Along with this, when the perceived efficacy of specific SAP(s) is high, the willingness to adopt the SAP would be greater. The related hypotheses are determined as follows:

**Hypothesis 2 (H2).** *Farmer's perception of the threat severity has a positive and significant effect on the adoption of SAP.*

**Hypothesis 3 (H3).** *Farmer's perception of the vulnerability to threat has a positive and significant effect on the adoption of SAP.*

**Hypothesis 4 (H4).** *Farmer's perceived self-efficacy has a positive and significant effect on the adoption of SAP.*

**Hypothesis 5 (H5).** *Farmer's perceived response-efficacy has a positive and significant effect on the adoption of SAP.*

Furthermore, many adoption-related studies have shown that information about the particular practice or training has a high impact on adoption [35]. Although the sources of information may vary, our study used an official source of information as an explanatory variable. It was defined as whether farmers had received information about SAP from the

Sustainable Agriculture Project framework, a joint German-Mongolian project. Thus, our last hypothesis is proposed as follows:

**Hypothesis 6 (H6).** *Obtaining information about SAP has a significant positive impact on the SAP adoption.*

Control variables such as experience, wheat acreage, and soil fertility are also employed to explain the adoption decision. Moreover, we are assuming a certain relationship between exogenous explanatory variables and perception variables. Soil fertility of a specific area could have a negative relationship between the farmer's perception of the threat and perception of vulnerability. In addition, the experience might have a significant effect on the farmers' perceived self-efficacy. These assumptions create indirect effects of soil fertility and experience on the adoption decision. The proposed conceptual model is presented in Figure 1.

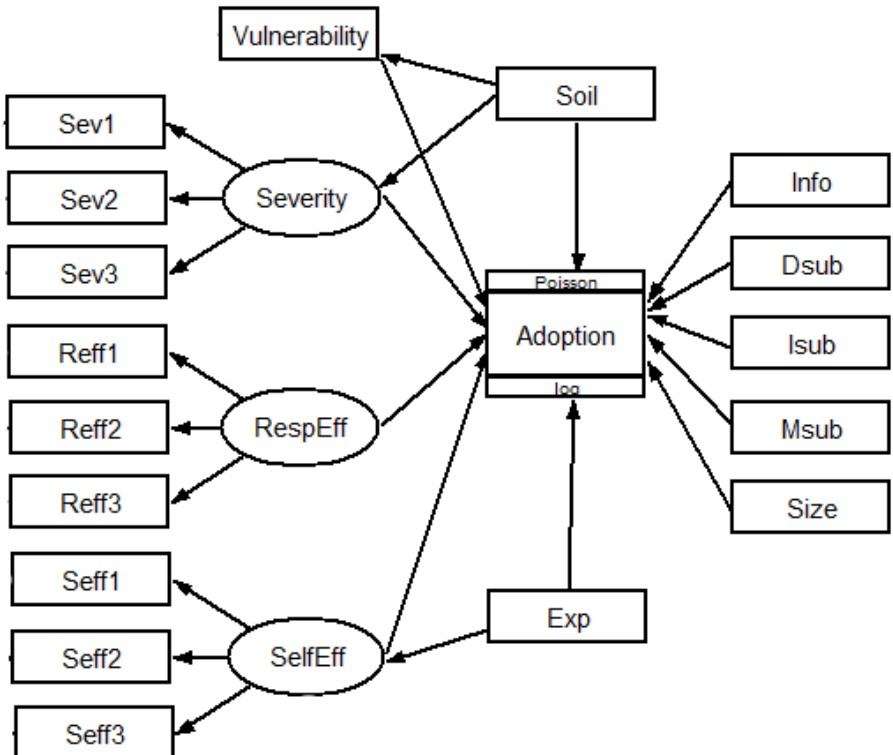

**Figure 1.** Conceptual framework (Error terms not included).

## 4. Materials and Methods

### 4.1. Analytical Methods

We assume a Poisson distribution for the dependent variable as it is the number of SAPs adopted, a count data. It is common to model the count data regression as a log-linear model [50]. The Poisson log-linear model with explanatory variable $X$ is:

$$log y_i = x_i'\beta, \tag{1}$$

where $y_i$ represents the number of sustainable practices adopted by the farmer (from 0 up to 4), and $x_i$ is the vector of explanatory variables, which include incentives, latent psychological constructs, and other controlling variables. $y_i$ has the following probability mass function:

$$p(y_i) = \frac{e^{-\mu_i}\mu_i^i}{y_i!}, \; y_i = 0,1,2,3,4. \tag{2}$$

Our conceptual framework includes latent constructs in the explanatory variables along with other observable covariates. Thus, to test our proposed hypothesis, both factor analysis and regression analysis/path analysis needed to be conducted simultaneously. For this purpose, this study employed the structural equation modeling (SEM) technique. More specifically, because the dependent variable is assumed to have Poisson distribution, generalized SEM (GSEM) was used. An SEM consists of two sub-models, a structural model representing the relationships between the latent constructs and a measurement model representing the relationships between the latent construct and their observable indicators [51]. Specifically, as demonstrated in [52]:

Measurement models:

$$y = \Lambda_y \eta + \varepsilon, \tag{3}$$

$$x = \Lambda_x \xi + \delta. \tag{4}$$

Structural model:

$$\eta = \beta\eta + \Gamma\xi + \zeta, \tag{5}$$

where $y$ is a vector of observed endogenous variables, $\Lambda_y$ is a matrix of latent construct loadings, and $\eta$ is a vector of latent endogenous variables. $x$ is a vector of observed exogenous variables, $\Lambda_x$ is a matrix of regression coefficients, $\xi$ is a vector of latent exogenous variables. $\beta$ represents the effect of the $j$th endogenous latent variable on the ith endogenous latent variable, whereas $\Gamma$ represents the effect of the $j$th exogenous latent variable on the $i$th endogenous latent variable. $\varepsilon$, $\delta$ and $\zeta$ are vectors of measurement errors. The conceptual model is shown in Figure 1 below.

According to SEM notation, our conception model in Figure 1 reads as follows. First, the endogenous measurement model, as in Equation (3):

$$
\begin{bmatrix}
Adoption \\
Sev\,1 \\
Sev\,2 \\
Sev\,3 \\
Seff\,1 \\
Seff\,2 \\
Seff\,3 \\
Reff\,1 \\
Reff\,2 \\
Reff\,3 \\
Vulnerability
\end{bmatrix}
=
\begin{bmatrix}
1 & 0 & 0 & 0 & 0 \\
0 & \lambda_{22} & 0 & 0 & 0 \\
0 & \lambda_{32} & 0 & 0 & 0 \\
0 & \lambda_{42} & 0 & 0 & 0 \\
0 & 0 & \lambda_{53} & 0 & 0 \\
0 & 0 & \lambda_{63} & 0 & 0 \\
0 & 0 & \lambda_{73} & 0 & 0 \\
0 & 0 & 0 & \lambda_{84} & 0 \\
0 & 0 & 0 & \lambda_{94} & 0 \\
0 & 0 & 0 & \lambda_{104} & 0 \\
0 & 0 & 0 & 0 & 1
\end{bmatrix}
\times
\begin{bmatrix}
Adoption \\
Severity \\
Self\ efficacy \\
Resp\ efficacy \\
Vulnerability
\end{bmatrix}
+
\begin{bmatrix}
\varepsilon_1 \\
\varepsilon_2 \\
\varepsilon_3 \\
\varepsilon_4 \\
\varepsilon_5
\end{bmatrix}. \tag{6}
$$

The exogenous measurement model, as in Equation (4):

$$
\begin{bmatrix}
Information \\
Support-direct \\
Support-input \\
Support-machinery \\
Land\ size \\
Experience \\
Soil\ condition
\end{bmatrix}
=
\begin{bmatrix}
1 & 0 & 0 & 0 & 0 & 0 & 0 \\
0 & 1 & 0 & 0 & 0 & 0 & 0 \\
0 & 0 & 1 & 0 & 0 & 0 & 0 \\
0 & 0 & 0 & 1 & 0 & 0 & 0 \\
0 & 0 & 0 & 0 & 1 & 0 & 0 \\
0 & 0 & 0 & 0 & 0 & 1 & 0 \\
0 & 0 & 0 & 0 & 0 & 0 & 1
\end{bmatrix}
\times
\begin{bmatrix}
Info \\
Dsup \\
Isup \\
Msup \\
Size \\
Exp \\
Soil
\end{bmatrix}. \tag{7}
$$

The structural model of our conceptual framework, as in Equation (5), is:

$$
\begin{bmatrix} Adoption \\ Severity \\ Self\ efficacy \\ Resp\ efficacy \\ Vulnerability \end{bmatrix} = \begin{bmatrix} 0 & \beta_{12} & \beta_{13} & \beta_{14} & \beta_{15} \\ 0 & 0 & 0 & 0 & 0 \\ 0 & 0 & 0 & 0 & 0 \\ 0 & 0 & 0 & 0 & 0 \\ 0 & 0 & 0 & 0 & 0 \end{bmatrix} \times \begin{bmatrix} Adop \\ Sev \\ Seff \\ Seff \\ Vul \end{bmatrix} + \begin{bmatrix} \gamma_{11} & \gamma_{12} & \gamma_{13} & \gamma_{14} & \gamma_{15} & \gamma_{16} & \gamma_{17} \\ 0 & 0 & 0 & 0 & 0 & 0 & \gamma_{27} \\ 0 & 0 & 0 & 0 & 0 & \gamma_{36} & 0 \\ 0 & 0 & 0 & 0 & 0 & 0 & 0 \\ 0 & 0 & 0 & 0 & 0 & 0 & \gamma_{57} \end{bmatrix} \times \begin{bmatrix} Info \\ Dsup \\ Isup \\ Msup \\ Size \\ Exp \\ Soil \end{bmatrix} + \begin{bmatrix} \varsigma_1 \\ \varsigma_2 \\ \varsigma_3 \\ \varsigma_4 \\ \varsigma_5 \end{bmatrix}. \quad (8)
$$

The use of GSEM allows all of these analyses to be estimated simultaneously along with taking count dependent variables into account, which is an advantage that is not available in any other method than the SEM technique [51]. In terms of software, we employed MPlus to estimate GSEM.

### 4.2. Study Area and Data Description

Mongolia is a landlocked country with a 1.5 million km$^2$ area, which is located between Russia and China. The arable land is classified into five general zones according to its heat and moisture properties: central, Khangai, western, eastern, and Gobi. The central region has favorable soil and climatic conditions for wheat growing, and it supplies about 70 percent of the total harvest [40].

The study area covered is shown in Figure 2. The survey covered 9 provinces in 4 agricultural regions (There are a total of 14 provinces that grow wheat, which has around 850 wheat farmers nationally. Because the farmers are sparsely settled in remote and vast areas, so the Gobi region has not been included due to time and cost factors). Farmers from each province were randomly selected for the sample. In total, 239 farmers responded to the pre-tested questionnaire survey. Further, 68% of the respondents were from the central region, while the number of farmers from the west, khangai, and east regions covered 8%, 11%, and 12%, respectively. The sample wheat production share in total wheat production of 2019 was around 26%.

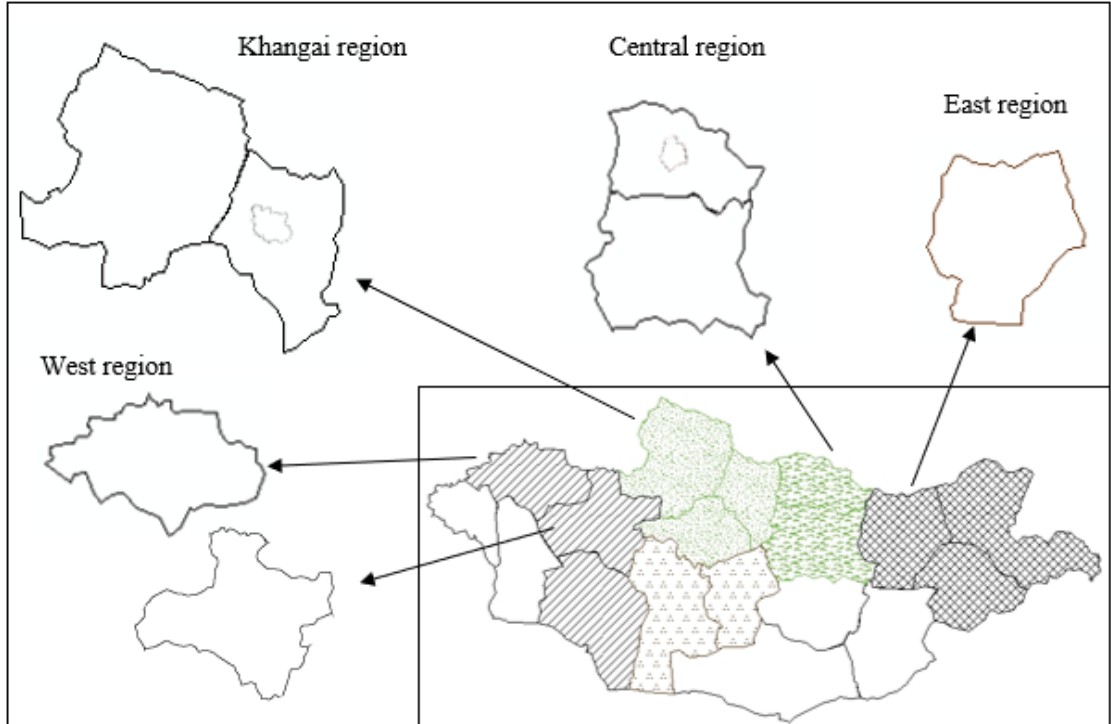

**Figure 2.** Study area—selected provinces from the regions.

Data were collected through pre-tested questionnaires via face to face interviews in the period between November and December of 2019. The questionnaire consisted of three parts: the part asking about crop production and farming practices; the part that asked about the government agricultural subsidy. The last part inquired about the perceptions related to soil erosion threat and SAPs.

The proportion of farmers using each practice was as follows. Among the farmers in the sample, 29% adopted the multi-cropping system and 76% use the reduced tillage system. In terms of organic fertilization, 72% of farmers use it for cultivation, while only 24 percent apply straw-mulching. In terms of the number of practices used, most of the farmers 49% used 2 types of SAPs and only 2.5% did not use any of them (Table 1). Descriptive statistics of variables except the perceptions are shown in Table 2.

**Table 1.** The proportion of farmers using a number of sustainable agriculture practices.

| Number of Practices Used | 0 | 1 | 2 | 3 | 4 |
|---|---|---|---|---|---|
| Proportion of farmers | 2.5% | 21.10% | 48.40% | 26.05% | 1.95% |

**Table 2.** Descriptive statistics.

| Variables | *Short Names* | Description | Mean | SD | Min | Max |
|---|---|---|---|---|---|---|
| | | Dependent variable | | | | |
| Intensity of adoption | *Adoption* | number of SAP-s adopted | 2.01 | 0.81 | 0 | 4 |
| | | Independent and control variables | | | | |
| Soil fertility | *Soil* | bonitet score fertility evaluation | 68.2 | 5.4 | 48.7 | 75.8 |
| Wheat acreage | *Size* | farming land, ha | 749.5 | 919.9 | 20 | 6648 |
| Experience | *Exp* | ordinal (1–5 scale: below 5 years above 20 years) | 3.64 | 1.3 | 1 | 5 |
| Information | *Info* | whether takes information from "sustainable agriculture" project (yes = 1, no = 0) | 0.56 | 0.5 | 0 | 1 |
| Direct subsidy | *Dsub* | number of times granted direct payment in the last 3 years | 1.93 | 1.19 | 0 | 3 |
| Input subsidy | *Isub* | number of times granted soft loans for inputs in the last 3 years | 0.79 | 0.88 | 0 | 3 |
| Investment subsidy | *Msub* | whether granted soft loans for agricultural machinery in the last 3 years | 0.33 | 0.47 | 0 | 1 |

The average adoption intensity is around 2, which implies the numbers of farmers who engaged in two kinds of SAPs are the most (49 percent of the farmers). Direct subsidy and input subsidy are measured as the number of times granted in the last three years. Forty-six percent of the farmers received direct payment on an annual basis, while 47 percent of the farmers had not received input support at all. The previous two subsidies were intended to be taken on an annual basis featuring short-term characteristics, while investment support had mid-term characteristics within 2 to 3 years. Thus, we defined this variable by whether farmers took this subsidy during the last 3 years.

Farmers do not measure the nutrients of their soil regularly, or at least not on a short-term basis, which makes farmers unable to know the precisely detailed nutritional information of their soil. Thus, we used a soil nutrient indicator rated on a bonitet scale point measured from every district (Soum) of each province reported by Enkhmaa [53]. This scale point reference is based on the geographical location, climate condition, soil characteristic.

According to the PMT framework, the following perception variables were used in the study (Table 3). Except for the perception of vulnerability to soil erosion, these psychological variables are treated as latent constructs, a variable that is not directly observable.

**Table 3.** Descriptive statistics for indicators of perceptions.

| Perceptions | *Short Names* | Indicators | Mean | SD | Min | Max |
|---|---|---|---|---|---|---|
| Perceptions of severity (fully disagree = 1; to fully agree = 5) | *Sev 1* | Soil erosion is getting worse over the past 3 years. | 3.30 | 0.85 | 1 | 5 |
| | *Sev 2* | Soil erosion is severe in my area. | 3.14 | 0.98 | 1 | 5 |
| | *Sev 3* | Soil erosion would be more severe in the coming 3 years if no measures were taken. | 3.71 | 0.79 | 1 | 5 |
| Perception of vulnerability (very low = 1; to very high = 5) | *Vulnerability* | The vulnerability of farmland to soil erosion. | 2.83 | 0.76 | 1 | 5 |
| Perception of self-efficacy (not at all = 1; completely = 5) | *Reff 1* | For me to use SAP is totally under my control. | 3.65 | 0.88 | 1 | 5 |
| | *Reff 2* | I have enough knowledge and competency for implementing SAP. | 3.25 | 1.03 | 1 | 5 |
| | *Reff 3* | I have sufficient resources financial, human, and technical for implementing SAP. | 2.90 | 0.84 | 1 | 5 |
| Perception of response-efficacy (fully disagree = 1; disagree = 2; somehow agree = 4; fully agree = 5) | *Seff 1* | Sustainable agricultural practices allow for the improvement of soil productivity. | 3.13 | 0.65 | 2 | 5 |
| | *Seff 2* | Sustainable agricultural practices allow for the improvement of economic benefits. | 2.66 | 0.78 | 1 | 5 |
| | *Seff 3* | Mentioned SAPs are effective ways to deal with the effects of soil erosion risks. | 2.71 | 0.72 | 1 | 4 |

## 5. Results and Discussion

### 5.1. Wheat Growing Regions and Intensity of Adoption

Our study covered farmers from four wheat-growing regions. The conceptual model did not include the region as an explanatory variable. The model already included the bonitet score, which represents the soil fertility. The region took into consideration in the estimation of the soil fertility score; thus, using these two variables at the same time can create multicollinearity. Therefore, we analyzed whether there were statistically significant differences in the adoption of intensity between the regions. Table 4 presents the mean adoption intensity and its standard deviation for each region.

**Table 4.** Summary of adoption intensity for regions.

| Regions | Mean | St. Dev | Frequency |
|---|---|---|---|
| Central | 1.99 | 0.82 | 163 |
| East | 2.10 | 0.72 | 29 |
| Khangai | 1.70 | 0.87 | 27 |
| Western | 2.50 | 0.61 | 20 |
| Total | 2.01 | 0.81 | 239 |

We conducted a one-way ANOVA to test the differences in farmers' adoption between the region. The result showed that the difference was statistically significant at the 1% significance level (Table 5).

**Table 5.** A one-way ANOVA result.

| Source | SS | Df | MS | F | *p*-Value |
|---|---|---|---|---|---|
| Between groups | 7.67 | 3 | 2.55 | 4.02 | 0.008 |
| Within groups | 149.29 | 235 | 0.63 | | |
| Total | 156.96 | 238 | 0.66 | | |
| Bartlett's test for equal variances: chi2(3) = 3.4039; Prob > chi2 = 0.333 | | | | | |

The ANOVA result suggests that the process of adopting and applying SAPs varies depending on the features of each region. For instance, ref. [19] studied land erosion in Mongolia between 1990 and 2015 and found that soil erosion rates vary from region to region. These findings can be used to formulate area-specific supportive policies for the further development of sustainable farming practices among the farmers.

### 5.2. Results of SEM

#### 5.2.1. Measurement Model—Reliability and Validity Test

The measurement model shows the relationship between the latent constructs and their indicators (Table 6). The standardized coefficients of the indicators are all significant and above the recommended value of 0.4, which shows good construct reliability, as stated in Henseler et al. [54]. Furthermore, the $R^2$ values are above the recommended level of 0.2 [52], except for the first indicator (self-efficacy 1) of self-efficacy perception.

**Table 6.** The measurement model (standardized coefficients).

| Latent Constructs | Indicators | Coefficient/Loadings/ | S.E | $R^2$ |
|---|---|---|---|---|
| Perception of severity | Severity 1 | 0.875 *** | 0.023 | 0.776 |
| | Severity 2 | 0.895 *** | 0.02 | 0.801 |
| | Severity 3 | 0.823 *** | 0.027 | 0.678 |
| Perception of self-efficacy | Self-efficacy 1 | 0.445 *** | 0.082 | 0.198 |
| | Self-efficacy 2 | 0.568 *** | 0.085 | 0.322 |
| | Self-efficacy 3 | 0.772 *** | 0.099 | 0.596 |
| Perception of response efficacy | Response efficacy 1 | 0.886 *** | 0.037 | 0.785 |
| | Response efficacy 2 | 0.776 *** | 0.045 | 0.602 |
| | Response efficacy 3 | 0.635 *** | 0.046 | 0.403 |

Note: *** Indicate that estimates are significant at the 1% probability level.

Since the coefficients are standardized, the reliability of the indicators of each construct can be compared. In terms of the perception of severity, the perception of the present is the most reliable indicator. When evaluating the perception of farmer's self-efficacy, self competency (self-efficacy 2) and available resources (self-efficacy 3) are the leading indicators. Moreover, the first indicator of response efficacy comes to the most reliable one, which means when the farmer has higher insights on the effectiveness of the SAP, they would consider the action will increase their productivity.

The test results of the reliability and validity of the latent constructs are shown in Table 7. The reliability tests, Cronbach's alpha, and the Construct reliability, are above the cut-off value of 0.7 except for perception of self-efficacy (In exploratory factor analysis the lower limit can be 0.6 (Robinson 1991). These reliability tests measure internal consistency, the degree to which a set of indicators of a latent construct is consistent in their measurements [51]. The validity test, average variance extracted (AVE), has also met the recommended level except for the second construct.

**Table 7.** Reliability and validity of perception constructs.

| Latent Constructs (Cut-Off Value) | AVE ($\geq$0.5) | Construct Reliability ($\geq$0.7) | Cronbach's ALPHA ($\geq$0.7) |
|---|---|---|---|
| Perception of severity | 0.85 | 0.94 | 0.90 |
| Perception of self-efficacy | 0.41 | 0.67 | 0.61 |
| Perception of response efficacy | 0.72 | 0.89 | 0.80 |

#### 5.2.2. The Results of the Structural Model

The ML estimation with robust standard errors (MLR) is used to estimate GSEM for examining the proposed hypotheses of this study. To the best of our knowledge, there are not any agreed-upon measures of model fit for GSEM yet. The conventional goodness fit

tests for SEM, such as RMSEA, CFI, or SRMR (where the acronyms are are the root mean square error of approximation (RMSEA), comparative fit index (CFI) and standardized root mean square residual (SRMR)), do not work for GSEM.

The structural model results are presented in Table 8. The standardized coefficients allow direct comparisons of the effects of the independent variables on the dependent variable [55]. In addition, the coefficients of log-linear regression are interpreted as the change in the log of the dependent variable given a one-unit change in the explanatory variable. This interpretation is not much illuminative; thus, using the incidence rate ratio (relative risk) to explain the coefficient would be more informative.

**Table 8.** The structural model (standardized coefficients).

| Variables | Adoption | Perception of Severity | Perception of Vulnerability | Perception of Self-Efficacy |
|---|---|---|---|---|
| Production subsidy | 0.013 (0.075) | | | |
| Input subsidy | 0.044 (0.067) | | | |
| Investment subsidy | 0.002 (0.068) | | | |
| Information | 0.563 *** (0.075) | | | |
| Soil fertility | −0.115 * (0.063) | −0.151 *** (0.062) | −0.012 (0.061) | |
| Wheat acreage | 0.080 (0.073) | | | |
| Experience | −0.093 (0.067) | | | −0.079 (0.077) |
| Vulnerability | 0.301 *** (0.08) | | | |
| Severity | 0.450 *** (0.077) | | | |
| Self-efficacy | 0.379 *** (0.071) | | | |
| Response efficacy | 0.419 *** (0.077) | | | |

Note: *** and * indicate that estimates are significant at the 1%and 10% probability levels, respectively. Standard errors are in parentheses.

The impacts of the subsidies on the adoption intensity were not significant, although their signs are as expected. Hence, the first hypothesis we had proposed was rejected. Whereas, at the beginning of the study, we had proposed that current government subsidies might motivate the farmers to operate efficiently in the long term and might encourage them to engage in sustainable farming activities. The result of the analysis is probably due to the fact these subsidies are not specified for encouraging SAPs. Previous studies have shown different relationships between government support and SAP adoption, including positive and negative, significant and not significant [8,15,35]. However, the supports considered in those studies differ from current research in that they are adoption payments to farmers for engaging in sustainable or conservation agricultural practices. Our study results show that to further develop the agricultural sector in a sustainable way, it is necessary to change the current subsidy policy and focus on encouraging farmers to use soil-friendly technologies.

The information had a significant positive effect. The coefficient expresses that the number of the SAPs adopted of the farmers who have taken information about SAPs was 75.6% (incidence rate, by exponentiating the coefficient) higher than those who have not. This result was consistent with the other adoption studies [35]. The wheat acreage has a positive

effect, though it is not significant (the coefficient of wheat acreage in Table 8). Apparently, the size of the wheat-growing land does not matter much for the adoption decision.

All the perception variables had significant and positive effects on the adoption intensity of SAPs, which are in line with our hypotheses. As assumed in the conceptual framework, a high level of appraisal of threats and response capabilities can stimulate adoption decisions. The results of the former studies, which included psychological variables, also concluded that farmers' perceptions of threats and the efficacy of the measures had a strong influence on adoption decisions [8,12,13,30,31,49].

In terms of sub-models of perceptions, the soil fertility evaluation score has a significant negative effect on perception severity. It appears that the farmers who are operating in fertile areas may feel less severe about soil erosion. The other coefficients of observed exogenous variables on the perceptions are not significant.

The estimated total effects of two explanatory variables are given in Table 9. The total effect on a dependent variable is the sum of its direct effect and its indirect effect via intervening variables [56]. Compared to their direct effects in Table 9, the total effects of these variables have a greater (in absolute quantity) and significant impact. The outcome shows that the farmers in the fertile areas tend to engage less in protecting or preventing measures. Experience had an adverse impact on adoption. An explanation might be that the farmers who have been engaged in farming for many years may have already begun to implement these practices or other alternatives before the time covered by the study.

**Table 9.** Estimated total effects (standardized coefficient).

| Variables | Coefficient | Errors |
|---|---|---|
| Soil fertility | −0.187 *** | 0.069 |
| Experience | −0.123 * | 0.072 |

Note: *** and * indicate that estimates are significant at the 1% and 10% probability levels, respectively.

The above results have implications for developing policies to motivate the farmers to use sustainable practices in their operation. Mongolia is in the transition to introduce sustainability in the development trend of agriculture; it is crucial to forming an optimal policy. As stated in [8], many countries use subsidies or economic incentives to affect the adoption decision. The Government of Mongolia also can alter their current subsidy policy in this regard. For instance, the cash payment basis can be related to the farmers' sustainable behavior. Another example can be that soft loans can be directed to the inputs and types of machinery used for sustainable farming. However, changes and modifications have to be conducted carefully as there are cases where economic incentives have not always been successful [57,58].

If a specific policy for motivating the adoption of SAPs is to be developed, the following aspects need to be considered. First, it is necessary to consider the specifics and differences of crop regions when developing policy. For instance, area-specific support policies could be more efficient (as resulted in 5–1). Furthermore, the policy should be specified differently depending on the scope of the farmer's activities (which can be defined by the size of the area under cultivation), but our study found that the wheat acreage did not have much effect on adoption decision. As the psychological variables and information are the significant determinants, increasing access to related information and raising farmers' awareness of environmental erosion can encourage sustainable farming behaviors among farmers and can magnify the effectiveness of any proposed policies.

## 6. Conclusions

To maintain agricultural productivity and achieve sustainability, encouraging farmers to use SAPs has been considered an essential goal in the development of agriculture. In this study, we examined the factors affecting the adoption of SAPs in the context of Mongolian wheat farmers. Our study fills in knowledge gaps in domestic research on sustainable farming behavior and incentive policies for agricultural production.

Farmers' adoption behavior was measured by adoption intensity as we assume it is appropriate to model sustainable behavior as a selection of multiple SAPs. The analytical approach we used in this study contributes to the existing research studies of adoption behavior by using generalized structural equation modeling (GSEM) with a count dependent variable (which has Poisson distribution) to analyze the impacts of observable and latent variables on adoption intensity.

Our findings show that current subsidies granted for wheat farmers do not have significant impacts on the adoption intensity of SAPs. This verifies that the current policy needs to be modified in order to develop sustainable agriculture. The psychological determinants based on protection motivation theory were found to have significant positive effects on adoption intensity. Specifically, farmers' perception toward the severity and vulnerability of soil erosion along with the farmer's appraisal of coping abilities of the SAPs and farmer's own ability to implement the SAPs can induce the farmers to conduct sustainable practices more. Many previous studies, covering different country contexts, have also identified the influence of farmers' psychological factors as essential determinants.

Furthermore, information regarding the SAPs was another key factor that leads to the adoption of these technologies. Together, all these outcomes of this study reveal policy implications for further development of agricultural support policy in order to develop sustainable farming. The study suggests that using economic supports, along with the measures to impact the farmers' perceptions and increase awareness, can lead to an appealing result for promoting the adoption of SAPs. Besides, because adoption intensity varied from region to region, further policies should be tailored to regional specifics rather than common to all areas.

The study is subject to several limitations. First, the study focused on the general impact of the current subsidy policy and other key determinants or controlling variables on the adoption decision of SAPs of the wheat farmers. In this case, the adoption of SAPs was taken as an indicator of farmers' sustainable behavior. Thus, detailed analysis or information on how the subsidy policy influenced soil fertility, which is the actual outcome of farming practices, was not covered in this study. Second, the study is limited to the protection motivation theory and several control variables in terms of possible determinants and controlling variables. Hence, the results may reflect omitted variable bias. Moreover, soil fertility condition, one control variable, was defined by the general fertility score of each district, not by the specific measure of each farmer's land, as this information was not available for all the farmers in the sample. Thus, the study can be upgraded by including more detailed variables that can affect adoption decisions. Finally, the sample size covered in this study is limited to around 30% of the total population and the results might cannot represent the whole situation. Nevertheless, this study can be a valuable starting point for further analysis in this matter.

Further research could focus on explaining the adoption of specific SAPs in order to get more insights for designing proper measures to promote the adoptions. As a result, useful information such as which kind of barriers and motivators for adopting certain sustainable practices are existing and how it can be solved or developed further by implementing specific policies and measures.

**Author Contributions:** Conceptualization, B.P. and D.O.; methodology, B.P.; software, B.P. and D.O.; validation, X.H., and X.X.; formal analysis, B.P.; investigation, B.P.; data curation, D.O.; writing—original draft preparation, B.P.; writing—review and editing, X.H. and X.X.; visualization, B.P.; supervision, X.H. and X.X.; All authors have read and agreed to the published version of the manuscript.

**Funding:** This research received no external funding.

**Institutional Review Board Statement:** Not applicable.

**Informed Consent Statement:** Not applicable.

**Data Availability Statement:** Data sharing not applicable.

**Acknowledgments:** We want to thank the Crop Supporting Fund, the implementing agency of the Ministry of Food, Agriculture and Light Industry of Mongolia, for providing information and assisting the primary data collection.

**Conflicts of Interest:** The authors declare no conflict of interest.

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
