# Peer review of "Farmer’s Perception, Agricultural Subsidies, and Adoption of Sustainable Agricultural Practices: A Case from Mongolia"

_sustainability, doi:10.3390/su13031524_

Round 1

Reviewer 1 Report

The paper fills an important gap in the research and contributes to highly significant issues for transitions to sustainability. The combination of factors that you investigate is original, and the approach may have relevance for research in other contexts. However, the initial sections require much more detail to serve the researchers in the field. Please provide details and statistics on wheat cultivation in this region, to give the reader a clear picture of the historical developments you mention, cultivated surfaces, yields, economic value and the number of people depending on this crop for their livelihoods. You talk about soil degradation, erosion and soil fertility without details: please be very specific about all these aspects, and include technical details (which may be offered in the sources you reference, but which should be available to the reader to follow your arguments). Please add a section about other factors that influence soil degradation and erosion in particular, such as topography, weather patterns, types of soil, presence or absence of trees and shrubs that retain water and reduce wind, etc. Some notes on climate projections for the region and implications for wheat farming would be welcome. Finally, since the question of subsidies is central in your study and your arguments, the reader needs details on the provisions of the various kinds of subsidies or other government support, including statistics. To maximise the benefits that readers can draw from your study, please also get support from a professional for editing of English language, which in your manuscript sometimes is clear, but at other times inadequate. Best of luck with your revisions!

Reviewer 2 Report

Abstract:

One should clearly state that governmental subsidies are not intended to promote sustainability. Please check consistence between: "government subsidies are found to have no effect" and "the current subsidy policy needs to be revised to not interact negatively with sustainable development in agriculture".

Introduction:

Erosion is only one of the processes/components affecting soil fertility (it is possible to have also soil salinisation, pollution, adverse selection of soil micro-organisms, etc), while in the paper perception of threats to soil fertility is related only to erosion processes (perhaps in Mongolia it is so, but it should be stated).

In the paragraph about Mongolia you speak about "soil depreciation" (using a term commonly used in Economics for describing the effects of wear and tear on fixed capitals) while in the remaining part of the paper the expression "soil degradation" is used. I wonder if there is a reason for that.

While the effects of government policy are quite clear (increase in production), it would be good to have more information on the parameters used to measure soil degradation, and on how it is possible to have increased production with declining soil fertility, the expected time-span to have negative impact on production and, if it is possible, the main causes of soil degradation. I would anticipate in this paragraph, as there is the need to better understand the effects of the current government policy on soil fertility (maybe in general, and not only in terms of SAP adoption). Also a short statement about the relations between soil fertility, vulnerability and perception of severity of erosion would help in better understanding the model.

As regard farmers' motivations to adopt sustainable practices, this paper focuses on sustainability in terms of capability for future production, i.e. on maintaining productivity in the long run, while there could be other issues at stake (such as water quality, food safety, biodiversity, etc.) related to other components of community wellbeing.

Adoption of SAPs

Row 122. Footnote "3" is missing

-----

I was wondering if, in the Mongolian context, all types of multi-crop rotation systems have the same impact on soil fertility and, again, if using manure or compost has the same effect. On overall, I would have been interested not only to know the intensity of SAP adoption (from 0 to 4 SAPs) but to have at least some qualitative comments on discussion/conclusions about the effects that those SAPs can have on preventing soil degradation (do they contribute in the exact way to this goal?).

Conceptual framework

About H1: It is possible that I am pessimistic about effects on sustainability determined by policies aiming to increase productivity, but I would have formulated the research question in the opposite way (Do Current agricultural subsidies have a negative impact on the SAP adoption?), which would also be more consistent with your hypothesis that in the last decades there was an increase in productivity, but also processes of soil degradation. Anyway, as you stress in the results, the coefficient is not significant, but is positive... 

H6. I would not substitute the term "significant" that has a precise statistic meaning with "strong", although this can seem to stress the importance of the effect.

row 229 and following: please, consider to change "land size" into "wheat acreage" or other name more related to the variable

Methodology...

row 242 and following. Consider revising as follows:

(1)

where yi represents the number if of sustainable practices adopted by the farmer (from 0 up to 4) and xi is the vector of explanatory variables, which include incentives, latent psychological constructs and other controlling variables.

yi has the following probability mass function

(2)

-------------------

Figure 1

I should move vulnerability on the left part of the graphic, in order to have all the endogenous/perception variables on the left hand side

It is quite difficult to fully understand the conceptual framework, since the list of variables, their meaning and the "short names" that have been used to designate them (e.g. "sev1, sev2, etc") could be deducted only by the analysis of the tables with the descriptive statistics, which are presented some pages later. It would be useful to include in the methodology a table providing information about the variables included in the analysis and the short names that have been used to designate them before presenting the conceptual model. Some more references about the variables which are been chosen / excluded, in comparison with previous researches could be useful.

Last but not least, if Soil fertility influences SAPs adoption, I would think that, in its turn, SAPs adoption has an impact on Soil fertility, so I would had a second arrow going in the opposite direction

Study area and data description

It would be useful if in the tables variable "short names" were added at the side of the full name. E.g. if "Seff" is the short for "Self efficacy" and "Reff" is the short for Response efficacy. I wonder what "(SAP 10)" at the end of "for me o use SAP is totally under my control" does mean.

------

Footnote 5. could you please indicate the number of farmers growing wheat in the 4 regions covered by the survey? And some info about the share of wheat production accounted by them?

------

Figure 2. the departure of the arrow indicating the lower left hand side area seems not to be correct, since the shape it is not that of the West region from which it departs, but of the lowest area in Khangai region. Please, check consistency

-------- 

Are perceptions (excluded vulnerability) measured with a Likert scale? The number of steps may be deducted by the minimum-maximum values in the table, but it would have been good to explicit it, e.g. from Fully disagree = 1, to fully agree = 5, since there are cases where the minimum and the maximum value are not reached (Reff1, minimum 2; Reff3, maximum 4).

Same thing for vulnerability for which I suppose that a scale "1.very low, 2. low, 3. medium, 4. high, 5. very high" has been used.

Results and discussion

Table 4: I would have added also data about "soil" and "perception of vulnerability". Are soil characteristics homogeneous within districts or considered as such (only an average bonitet score for each district)?

-------

Table 5. It would be good for the sake of clarity to specify on which variables the ANOVA was performed.

--------

row 351. Actually, 0.198 (see Self efficacy 1 R2 value in table 6) is less than 0.2

--------

Please, ensure that all acronyms are explained either in text or in footnotes (e.g. RMSEA, CFI, SRMR) or put a reference. 

--------

row 400. Please change "insignificant" with "not significant"

--------

row 422. Another could be that soil that are still fertile after some decades of high productivity practices is quite stable and able to retain its fertility also in those conditions (low vulnerability). I don't know if this hypothesis could be realistic in the areas under analysis

-----------

442 Target policies promoting SAPs by taking into account for each SAP in each Region both the cost of adoption by farmers and the effect (and its perception) on soil fertility, in order to improve efficiency of funding. For instance, reduced tillage SAP seems to be already well spread (76% of farmers already adopt them) as well the use of manure and compost, while straw mulching and multi-cropping systems are less "popular". This means that either the unpopular SAPs are not considered to contribute to the protection of soil fertility, or they are too "costly" (giving a broad meaning to the term, e.g. costly from an economic viewpoint, but also in terms of required competency, etc.). Reduced tillage may have appeal for reducing erosion while reducing time and labour required for soil processing, thus being profitable for present economic reason. In some way, while the paper explores SAPs relations with long-term productivity (due to the maintenance or improvement of soil fertility) it does not take into account the possibility that these practices are also economically profitable, at least in some farm typology (e.g. depending on farm acreage), in the short term.

Conclusions

As future prosecution of the research, maybe it would be interesting to understand why some SAPs are quite "popular" while other are not, especially if those, as it seems from the paper, have good potentiality in improving soil fertility. Once understood the reason, it would be easier to design proper measure to improve them (incentives, extension services providing information, etc.)

Reviewer 3 Report

The conception of paper is good and relevant. Also, the hypothesis' are well formulated.

The paper needs to add a literature review chapter. In this chapter Authors should add the improved literature review section separated from the introduction. here they should add more literature, especially from international peer-reviewed Journals.

It's worth to add limitation of the paper.

It would be good to put the aim of the study also in abstract.
